# SCOP: Scientific Control for Reliable Neural Network Pruning

**Yehui Tang**[1,2], **Yunhe Wang**[2], **Yixing Xu**[2], **Dacheng Tao**[3],
**Chunjing Xu**[2], **Chao Xu**[1], **Chang Xu**[3]
[1]Key Lab of Machine Perception (MOE), Dept. of Machine Intelligence, Peking University.
[2]Noah's Ark Lab, Huawei Technologies.
[3]School of Computer Science, Faculty of Engineering, University of Sydney.
yhtang@pku.edu.cn, {yunhe.wang, yixing.xu, xuchunjing}@huawei.com,
{dacheng.tao,c.xu}@sydney.edu.au, xuchao@cis.pku.edu.cn.

## Abstract

This paper proposes a reliable neural network pruning algorithm by setting up a scientific control. Existing pruning methods have developed various hypotheses to approximate the importance of filters to the network and then execute filter pruning accordingly. To increase the reliability of the results, we prefer to have a more rigorous research design by including a scientific control group as an essential part to minimize the effect of all factors except the association between the filter and expected network output. Acting as a control group, knockoff feature is generated to mimic the feature map produced by the network filter, but they are conditionally independent of the example label given the real feature map. We theoretically suggest that the knockoff condition can be approximately preserved given the information propagation of network layers. Besides the real feature map on an intermediate layer, the corresponding knockoff feature is brought in as another auxiliary input signal for the subsequent layers. Redundant filters can be discovered in the adversarial process of different features. Through experiments, we demonstrate the superiority of the proposed algorithm over state-of-the-art methods. For example, our method can reduce 57.8% parameters and 60.2% FLOPs of ResNet-101 with only 0.01% top-1 accuracy loss on ImageNet. The code is available at `https://github.com/huawei-noah/Pruning/tree/master/SCOP_NeurIPS2020`.

## 1 Introduction

Convolutional neural networks (CNNs) have been widely used and achieve great success on massive computer vision applications such as image classification [18, 12, 38, 32], object detection [29, 43, 35] and video analysis [36]. However, due to the high demands on computing power and memory, it is hard to deploy these CNNs on edge devices, *e.g.*, mobile phones and wearable gadgets. Thus, many algorithms have been recently developed for compressing and accelerating pre-trained networks including quantization [7, 31, 39], low-rank approximation [19, 40, 42], knowledge distillation [16, 37, 41, 6], network pruning [27, 2, 34] *etc*.

Based on the different motivations and strategies, network pruning can be divided into two categories, *i.e.*, weight pruning and filter pruning. Weight pruning aims to eliminate weight values dispersedly, and the filter pruning removes the entire redundant filters. The latter has been paid much attention to as it can achieve practical acceleration without specific software and hardware design. Specifically, the redundant filters will be directly eliminated for establishing a more compact architecture with similar performance.

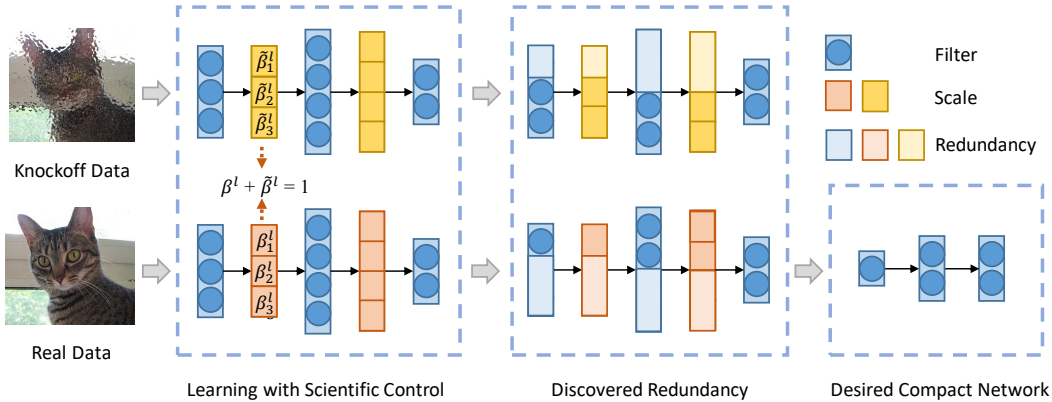

Figure 1: Diagram of the proposed filter pruning method with scientific control (SCOP). The real data and their knockoffs are simultaneously inputted to the pre-trained network. The knockoff features are seen as the control group to help discover redundancy in real features. Filters with large scaling factors for knockoff features but small factors for real features should be pruned.

The most important component for filter pruning is how to define the importance of filters, and unimportant filters can be removed without affecting the performance of pre-trained networks. For example, a typical hypothesis is 'smaller-norm-less-important' and filters with smaller norms will be pruned [20, 9]. Besides, He *et al.* [10] believed that filters closest to the 'geometric median' are redundant and should be pruned. Considering the input data, Molchanov *et al.* [27] estimated the association between filters and the final loss with Taylor expansion and preserved filters with close association. From an optimization viewpoint, Zhuo *et al.* [46] introduced a cogradient descent algorithm to make neural networks sparse, which provided the first attempt to decouple the hidden variables related to pruning masks and kernels. Instead of directly discarding partial filters, Tang *et al.* [33] proposed to develop new compact filters via refining information from all the original filters.

However, massive potential factors are inevitably introduced when developing a specific hypothesis to measure the importance of filters, which may disturb the pruning procedure. For example, the dependence between different channels may mislead norm based methods as some features/filters containing no useful information but have large norms. For some data-driven methods, the importance ranking of filters may be sensitive to slight changes of input data, which incurs unstable pruning results. Actually, more potential factors are accompanied by specific pruning methods and also depend on different scenarios, it is challenging to enumerate and analyze them singly when designing a pruning method.

In this paper, we propose a reliable network pruning method by setting up a scientific control for reducing the disturbance of all the potential irrelevant factors simultaneously. *Knockoff* features, having a similar distribution with real features but independent with ground-truth labels, are used as the control group to help excavate redundant filters. Specifically, knockoff data are first generated and then fed to the given network together with the real training dataset. We theoretically prove that the intermediate features from knockoff data can still be seen as the knockoff counterparts of features from real data. The two groups of features are mixed with learnable scaling factors and then used for the next layer during training. Filters with larger scaling factors for knockoffs should be pruned, as shown in Figure 1. Experimental results on benchmark models and datasets illustrate the effectiveness of the proposed filter pruning method under the scientific control rules. The models obtained by our approach can achieve higher performance with similar compression/acceleration ratios compared with the state-of-the-art methods.

## 2    Preliminaries and Motivation

In this section, we firstly revisit filter pruning in deep neural networks in the perspective of feature selection, and then illustrate the motivation of utilizing the generated knockoffs as the scientific control group for excavating redundancy in pre-trained networks.

Filter pruning is to remove redundant convolution filters from pre-trained deep neural networks for generating compact models with lower memory usage and computational costs. Given a CNN $f$ with $L$ layers, filters in the $l$-th convolutional layer is denoted as $\mathcal{W}^l = \{\boldsymbol{w}_1^l, \boldsymbol{w}_2^l, \cdots, \boldsymbol{w}_{M^l}^l\}$, in which $M^l$ is the number of filters in the $l$-th layer. $\boldsymbol{w}_i^l \in \mathbb{R}^{C^l \times k \times k}$ is a filter with $C^l$ input channels and kernel size $k$. A general objective function of the filter pruning task [20, 10] for the $l$-th layer can be formulated as:

$$\min_{\mathcal{W}^l} E(X, \mathcal{W}^l, Y), \quad s.t. \quad ||\mathcal{W}^l||_0 \leq \kappa^l, \tag{1}$$

where $X$ is the input data and $Y$ is the ground-truth labels, $E(\cdot, \cdot)$ is the loss function related to the given task (*e.g.*, cross-entropy loss for image classification), $|| \cdot ||_0$ is $\ell_0$-norm representing the number of non-zero elements and $\kappa^l$ is the desired number of preserved filters for the $l$-th layer.

Filters themselves have no explicit relationship with the data, but features produced by filters with the given input data under the supervision of labels can well reveal the importance of the filters in the network. In contrast with the direct investigation over the network filters in Eq. (1), we focus on features produced by filters in the pre-trained network. Eq. (1) can thus be reformulated as:

$$\min_{\mathcal{A}^l} E(\mathcal{A}^l, Y), \quad s.t. \quad ||\mathcal{A}^l||_0 \leq \kappa^l, \tag{2}$$

where $\mathcal{A}^l = f^l(X, \mathcal{W}^{1:l})$ is the feature in the $l$-th layer. Thus, the purpose of filter pruning is equivalent to exploring redundant features produced by the original heavy network. The key of a feature selection procedure is to accurately discover the association between features $\mathcal{A}^l$ and ground-truth labels $Y$, and preserve features that are critical to the prediction. The filters corresponding to those selected features are then the more important ones and should be preserved after pruning.

Deriving the features that are truly related to the example labels through Eq. (2) is a nontrivial task, as there could be many annoying factors that affect our judgment, *e.g.*, interdependence between features, the fluctuation of input data and those factors accompanied with specific pruning methods. To reduce the disturbance of irrelevant factors, we set up a scientific control. Taking the real features produced by the network filters as the treatment group for excavating redundancy, we generate their knockoff counterparts as the control group, so as to minimize the effect of irrelevant factors. Knockoff counterparts are defined as follows:

**Definition 1** *Given the real feature $\mathcal{A}^l$, the knockoff counterpart $\tilde{\mathcal{A}}^l$ is defined as a random feature with the same shape of $\mathcal{A}^l$, which satisfies the following two properties,* i.e., *exchangeability and independence [1]:*

$$[\mathcal{A}^l, \tilde{\mathcal{A}}^l] \overset{dis.}{=} [\mathcal{A}^l, \tilde{\mathcal{A}}^l]_{swap(\tilde{S})}, \tag{3}$$

$$\tilde{\mathcal{A}}^l \perp\!\!\!\perp Y | \mathcal{A}^l, \tag{4}$$

*for all $\tilde{S} \subseteq \{1, 2, \cdots, d\}$, where $\overset{dis.}{=}$ denotes equation on distribution, $Y$ denotes the corresponding labels and $d$ is the number of elements in $\mathcal{A}^l$. $[\cdot, \cdot]$ denotes concatenating two features and the $[\mathcal{A}^l, \tilde{\mathcal{A}}^l]_{swap(\tilde{S})}$ is to swap the $j$-th element in $\mathcal{A}^l$ with that in $\tilde{\mathcal{A}}^l$ for each $j \in \tilde{S}$.*

As shown in Eq. (3), swapping the corresponding elements between the real feature and its knockoff does not change the joint distribution. The knockoff counterparts always behave similar to real features, and hence are equally affected by those potential irrelevant factors. Whereas, the knockoff feature is conditionally independent with the corresponding label given real feature (Eq. (4)), containing no information about ground-truth labels. Thus, the only difference between them is that real features may have an association with labels while knockoffs do not. Through comparing the effects of real features and their knockoff counterparts in the network, we can then minimize the disturbance of irrelevant factors and force the selection procedure to only focus on the association between features and predictions, which leads to a more reliable way to determine the importance of filters.

## 3 Approach

In this section, we introduce how to construct effective knockoff counterparts and then illustrate the proposed reliable network filter pruning algorithm.

## 3.1 Knockoff Data and Features

Given the real features, their knockoff counterparts can be constructed with a generator $\mathcal{G}$, *i.e.*, $\tilde{\mathcal{A}}^l = \mathcal{G}(\mathcal{A}^l)$. There are a few approaches to generate quality knockoffs satisfying Definition 1. For example, Candes *et al.* [1] constructed approximate knockoff counterparts by analyzing the statistic properties of the given features, and Jordon *et al.* [13] generated knockoffs via a deep generative model. These methods can generate quality knockoffs for the given features, but it would be challenging for them to repeat the training of knockoff generating models for each individual layer in a deep neural network to be pruned.

By investigating the information flow in the neural network, we plan to develop a more efficient method to generate knockoff features for the network pruning task. We theoretically analyze the features generated from the real and knockoff data, and prove that the knockoff condition (Definition 1) can be approximately preserved. Thus, we only need to generate the knockoff counterparts of input data and efficiently derive knockoff features in all layers.

Specifically, we divide the components of the neural networks into two categories, nonlinear activation layers (*e.g.*, ReLU, Sigmoid) and linear transformation layers (*e.g.*, full-connected layers, convolutional layers), and give the proof respectively. For the activation layers, we have the following lemma.

**Lemma 1** *Suppose $\tilde{\mathcal{A}}^l$ is the knockoff counterparts of the input feature $\mathcal{A}^l$, the corresponding output features $\phi(\tilde{\mathcal{A}}^l)$ is still the knockoff of $\phi(\mathcal{A}^l)$, in which $\phi$ denotes any element-wise activation function.*

As element-wise operation does not change the dependence between different elements in a joint distribution, $\phi(\mathcal{A}^l)$ and $\phi(\tilde{\mathcal{A}}^l)$ still satisfy the exchangeability property (Eq. (3)). The independence property (Eq. (4)) is also satisfied as the activation functions do not introduce any information about labels. Thus Lemma 1 holds and the knockoff condition can be preserved across activation layers.

In linear transformation layers, we denote $\boldsymbol{v}^l \in \mathbb{R}^{1 \times d^l}$ as the vectorization of feature $\mathcal{A}^l$ and $\tilde{\boldsymbol{v}}^l$ for $\tilde{\mathcal{A}}^l$ for symbol simplicity. The linear transformation is denoted as $t(\boldsymbol{v}^l) = \boldsymbol{v}^l W$, where $W \in \mathbb{R}^{d^l \times d^{l+1}}$ is the transform matrix[1]. Instead of directly verifying the distribution of $[\boldsymbol{v}^l, \tilde{\boldsymbol{v}}^l]$ and $[\boldsymbol{v}^l, \tilde{\boldsymbol{v}}^l]_{swap(\tilde{S})}$ for arbitrary subset $\tilde{S}$, we check whether the two distributions have the same moments. In general, a higher moment means more precise analysis but also with more complexity. Many works compared the first two moments (*i.e.*, expectation and covariance) of distributions and obtain reliable conclusions in practice [14, 1]. Following them, we also focus on the first two moments. Given $\tilde{\boldsymbol{v}}^l \perp\!\!\!\perp Y | \boldsymbol{v}^l$, we denote $\tilde{\boldsymbol{v}}^l$ as the second-order knockoff of $\boldsymbol{v}^l$ if the expectation & covariance of $[\boldsymbol{v}^l, \tilde{\boldsymbol{v}}^l]$ and $[\boldsymbol{v}^l, \tilde{\boldsymbol{v}}^l]_{swap(\tilde{S})}$ are the same, and then have the following lemma:

**Lemma 2** *Suppose $\tilde{\boldsymbol{v}}^l$ is the second-order knockoff counterpart of input feature $\boldsymbol{v}^l$, the corresponding output feature $\tilde{\boldsymbol{v}}^{l+1} = t(\tilde{\boldsymbol{v}}^l) + \tilde{\boldsymbol{b}}^l$ is still the second-order knockoff of $\boldsymbol{v}^{l+1} = t(\boldsymbol{v}^l) + \boldsymbol{b}^l$, where bias $\boldsymbol{b}^l$ and $\tilde{\boldsymbol{b}}^l$ are random variables with zero means, and the covariance matrix of the joint distribution $[\boldsymbol{b}^l, \tilde{\boldsymbol{b}}^l]$ satisfies:*

$$\text{cov}[\boldsymbol{b}^l, \tilde{\boldsymbol{b}}^l] = \left[ \begin{array}{cc} \Sigma_{\boldsymbol{b}^l} & \Sigma_{\boldsymbol{b}^l} + W^T \text{diag}\{s^l\}W - diag\{s^{l+1}\} \\ \Sigma_{\boldsymbol{b}^l} + W^T diag\{s^l\}W - \text{diag}\{s^{l+1}\} & \Sigma_{\boldsymbol{b}^l} \end{array} \right], \quad (5)$$

*where $\boldsymbol{b}^l$, $\tilde{\boldsymbol{b}}^l$ have the same covariance $\Sigma_{\boldsymbol{b}^l}$ and $diag\{s^l\}$, $diag\{s^{l+1}\}$ are diagonal matrices.*

The proof is shown in the supplemental material. Lemma 2 shows that the knockoff condition can be approximately preserved across a linear transformation layer, with biases $\boldsymbol{b}^l$ and $\tilde{\boldsymbol{b}}^l$ as modified terms. In Eq. (5), $\text{diag}\{s^l\}$ depends on the previous layer while $\Sigma_{\boldsymbol{b}^l}$ and $\text{diag}\{s^{l+1}\}$ can be arbitrarily appointed as long as the covariance matrix is positive semi-definite. Given the mean and covariance, these biases can be directly generated to modify the outputs of convolutional layers in the procedure of excavating redundant filters. We also empirically investigate the effect of the biases in the ablation studies (Section 4.3). Combining Lemma 1 and Lemma 2, we then have the following conclusion:

**Proposition 1** *Suppose $\tilde{X}$ is the knockoff counterpart of the input data $X$, the corresponding feature $\tilde{\mathcal{A}}^l = f^l(\tilde{X}, \mathcal{W}^{1:l})$ in any layer can be approximately seen as the knockoff counterpart of the real feature $\mathcal{A}^l = f^l(X, \mathcal{W}^{1:l})$.*

Proposition 1 holds for general deep neural networks, and it provides much convenience to generating knockoffs for features in CNNs. We only need to generate knockoff counterparts of samples in the real training dataset, *i.e.*, $\tilde{X} = \mathcal{G}(X)$. Then the knockoff data are fed to the given network to derive knockoff features in all the layers, *i.e.*, $\tilde{A}^l = f^l(\tilde{X}, W^{1:l})$. We generate the knockoff data via a deep generative model and the details can be found in the supplemental material. Note that the complexity of generating knockoffs of the real dataset can be ignored, as they only need to be generated once and then utilized for all the tasks on a given dataset.

## 3.2 Filter Pruning with Scientific Control

In this section, we discuss the way of utilizing the generated knockoff features $\tilde{\mathcal{A}}^l$ as the control group, and then excavate redundant filters in neural networks. We put the knockoff feature $\tilde{\mathcal{A}}^l = f^l(\tilde{X}, \mathcal{W}^{1:l})$ together with its corresponding real feature $\mathcal{A}^l = f^l(X, \mathcal{W}^{1:l})$ as the input to the $(l+1)$-th layer, and a selection procedure is designed to select relevant features from them, *i.e.*,

$$\min_{\mathcal{A}^l, \tilde{\mathcal{A}}^l} E([\mathcal{A}^l, \tilde{\mathcal{A}}^l], Y), \quad s.t. \quad ||[\mathcal{A}^l, \tilde{\mathcal{A}}^l]||_0 \leq \kappa^l, \tag{6}$$

where $E$ is the loss function and $Y$ denotes ground-truth labels. The real feature $\mathcal{A}^l$ and its knockoff counterpart $\tilde{\mathcal{A}}^l$ are the responses of the network on real and knockoff data respectively, but knockoffs contain no information about labels. Here the real feature $\mathcal{A}^l$ is seen as the treatment group, whose relation with the network output is to be justified, while the knockoff features act as the control group to minimize the effect of potential irrelevant factors.

To implement the selection procedure (Eq. (6)) for a pre-trained deep neural network, we insert an adversarial selection layer after each convolutional layer in the network, *i.e.*,

$$\mathcal{A}^{l+1} = \phi(\mathcal{W}^{l+1} * (\boldsymbol{\beta}^l \odot \mathcal{A}^l + \tilde{\boldsymbol{\beta}}^l \odot \tilde{\mathcal{A}}^l)), \tag{7}$$

where $\boldsymbol{\beta}^l \in [0, 1]^{M^l}$ and $\tilde{\boldsymbol{\beta}}^l \in [0, 1]^{M^l}$ are the scaling factors of the real feature and knockoff feature, with constraint $\boldsymbol{\beta}^l + \tilde{\boldsymbol{\beta}}^l = \mathbf{1}$. $\phi$ is the activation function, $*$ is the convolutional operation and $\odot$ denotes element-wise multiplication. Besides the real features, knockoff features also have an opportunity to participate in the calculation of the subsequent layers, which depends on the scaling factors. The two groups of features compete with each other and efficient filters can be discovered in the adversarial process.

As analyzed in Section 3.1, the knockoff features $\tilde{\mathcal{A}}^l$ can be obtained by using the knockoff data $\tilde{X}$ as input, *i.e.*, $\tilde{\mathcal{A}}^l = f^l(\tilde{X}, \mathcal{W}^{1:l})$. In practice, we can get the real feature $\mathcal{A}^l$ and their knockoff counterparts $\tilde{\mathcal{A}}^l$ by feeding $X$ and $\tilde{X}$ to the network, simultaneously. Thus the scaling factors $\boldsymbol{\beta}^l$ and $\tilde{\boldsymbol{\beta}}^l$ can be optimized under the supervision of labels $Y$, by taking both real data $X^l$ and their knockoffs $\tilde{X}^l = \mathcal{G}(X)$ as input, *i.e.*,

$$\min_{\boldsymbol{\beta}, \tilde{\boldsymbol{\beta}}} E(X, \tilde{X}, Y, \boldsymbol{\beta}, \tilde{\boldsymbol{\beta}}), \quad s.t. \quad \boldsymbol{\beta}^l + \tilde{\boldsymbol{\beta}}^l = \mathbf{1}, l \in [1, 2, \cdots, L], \tag{8}$$

where $\boldsymbol{\beta}$ and $\tilde{\boldsymbol{\beta}}$ denote all the scaling factors in the whole network. During optimization, the parameters of the pre-trained network (*e.g.*, weights in convolutional layers) are fixed and only the scaling factors $\boldsymbol{\beta}^l, \tilde{\boldsymbol{\beta}}^l$ are updated to excavate redundant filters.

After minimizing Eq. (8), the obtained scaling factors $\boldsymbol{\beta}^l, \tilde{\boldsymbol{\beta}}^l$ can be used to measure the importance of features, *i.e.*, a feature with a large control scale means it is considered as important by the selection procedure. A filter $\boldsymbol{w}_j^l \in \mathcal{W}^l$ produces both the real and knockoff features while the knockoff contains no information about labels. Intuitively, if the real feature cannot suppress its knockoff counterpart (*i.e.*, small $\beta_j^l$ and large $\tilde{\beta}_j^l$), the corresponding filter $\boldsymbol{w}_j^l$ should be pruned. Thus a statistic $\boldsymbol{\mathcal{I}}^l = \boldsymbol{\beta}^l - \tilde{\boldsymbol{\beta}}^l$ is defined to measure the importance of each filter $\boldsymbol{w}_j^l \in \mathcal{W}^l$.[2] With a given pruning

Table 1: Comparison of the pruned networks with different methods on CIFAR-10. 'Orig.'/'Pruned' denote the test errors of the pre-trained/pruned networks, and 'Gap' is their difference. 'Params.' ↓ (%) and 'FLOPs' ↓ (%) are reduced percentages of parameters and FLOPs, respectively.

| Model | Method | Error (%) | | | Params. ↓ (%) | FLOPs ↓ (%) |
| --- | --- | --- | --- | --- | --- | --- |
| | | Original | Pruned | Gap | | |
| ResNet-20 | SFP (2018) [9] | 7.80 | 9.17 | 1.37 | 39.9 | 42.2 |
| | FPGM (2019) [10] | 7.80 | 9.56 | 1.76 | 51.0 | 54.0 |
| | SCOP (Ours) | 7.78 | **9.25** | **1.44** | **56.3** | **55.7** |
| ResNet-32 | MIL (2017) [5] | 7.67 | 9.26 | 1.59 | N/A | 31.2 |
| | SFP (2018) [9] | 7.37 | 7.92 | 0.55 | 39.7 | 41.5 |
| | FPGM (2019) [10] | 7.37 | 8.07 | 0.70 | 50.8 | 53.2 |
| | SCOP (Ours) | 7.34 | **7.87** | **0.53** | **56.2** | **55.8** |
| ResNet-56 | CP (2017) [11] | 7.20 | 8.20 | 1.00 | N/A | 50.0 |
| | SFP (2018) [9] | 6.41 | 7.74 | 1.33 | 50.6 | 52.6 |
| | GAL (2019) [23] | 6.74 | 7.26 | 0.52 | 44.8 | 48.5 |
| | FPGM (2019) [10] | 6.41 | 6.51 | 0.10 | 50.6 | 52.6 |
| | HRank (2020) [22] | 6.74 | 6.83 | 0.09 | 42.4 | 50.0 |
| | SCOP (Ours) | 6.30 | **6.36** | **0.06** | **56.3** | **56.0** |
| MobileNetV2 | DCP (2018) [45] | 5.53 | 5.98 | 0.45 | 23.6 | 26.4 |
| | SCOP (Ours) | 5.52 | **5.76** | **0.24** | **36.1** | **40.3** |

rate, filter $w_j^l$ with smaller $\mathcal{I}_j^l \in \mathcal{I}^l$ is pruned to get a compact network. The preserved filters can be reliably considered as having close association with expected network output, as other potential factors are minimized via the control group. At last, the pruned network is fine-tuned to further improve the performance.

# 4 Experiments

In this section, we empirically investigate the proposed filter pruning method (SCOP) by extensive experiments on benchmark dataset CIFAR-10 [17] and large-scale ImageNet (ILSVRC-2012) dataset [3]. CIFAR-10 dataset contains 60K RGB images from 10 classes, 50K images for training and 10K for testing. Imagenet (ILSVRC-2012) is a large-scale dataset containing 1.28M training images and 50K validation images from 1000 classes. ResNet [8] with different depths and light-weighted MobilenetV2 [30] are pruned to verify the effectiveness of the proposed method. The pruned models have been included in the MindSpore model zoo [3].

**Implementation details.** For the pruning setting, all the layers are pruned with the same pruning rate following [9] for a fair comparison. Based on the pre-trained model, the scaling factors $\beta^l$ and $\tilde{\beta}^l$ are optimized with Adam [15] optimizer, while all other parameters in the network are fixed. On CIFAR-10 dataset, the learning rate, batchsize and the number of epochs are set to 0.001, 128 and 50, while those on ImageNet are 0.004, 1024, and 20. The initial value of scaling factors are set to 0.5 for a fair competition between the treatment and control groups. The pruned network is then fine-tuned for 400 epochs on CIFAR-10 and 120 epochs on ImageNet, while the initial learning rates are set to 0.04 and 0.2, respectively. Standard data augmentations containing random crop and horizontal flipping are used in the training phase. The experiments are conducted with Pytorch [28] and MindSpore [4] on NVIDIA V100 GPUs.

## 4.1 Comparison on CIFAR-10

The comparison of different methods on CIFAR-10 is shown in Table 1. The pruning rate of the proposed method is set to 45%. SFP [9],FPGM [10]and Hrank [22] are SOTA filter pruning methods, measuring the importance of filters via norm, 'geometric median' and the rank of feature maps, respectively. Compared to them, our method achieves lower test error while more parameters and FLOPs are reduced. For example, our method achieves 6.36% error (0.06% accuracy drop) after pruning 56.0% FLOPs of ResNet-56, which are better than other methods such as HRank [10](6.83%

Table 2: Comparison of the pruned ResNet with different methods on ImageNet (ILSVRC-2012). 'Orig.'/'Pruned' denote the test errors of the pre-trained/pruned networks, and 'Gap' is their difference. 'Params.' ↓ (%) and 'FLOPs' ↓ (%) are reduced percentages of parameters and FLOPs, respectively.

| Model | Method | Top-1 Error (%) | | | Top-5 Error (%) | | | Params. | FLOPs |
| | | Orig. | Pruned | Gap | Orig. | Pruned | Gap | ↓ (%) | ↓ (%) |
|---|---|---|---|---|---|---|---|---|---|
| Res18 | MIL (2017) [5] | 30.02 | 33.67 | 3.65 | 10.76 | 13.06 | 2.30 | N/A | 33.3 |
| | SFP (2018) [9] | 29.72 | 32.90 | 3.18 | 10.37 | 12.22 | 1.85 | 39.3 | 41.8 |
| | FPGM (2019) [10] | 29.72 | 31.59 | 1.87 | 10.37 | 11.52 | 1.15 | 39.3 | 41.8 |
| | PFP-A (2020) [21] | 30.26 | 32.62 | 2.36 | 10.93 | 12.09 | 1.16 | 43.8 | 29.3 |
| | PFP-B (2020) [21] | 30.26 | 34.35 | 4.09 | 10.93 | 13.25 | 2.32 | 60.5 | 43.1 |
| | SCOP-A (Ours) | 30.24 | **30.82** | **0.58** | 10.92 | **11.11** | **0.19** | 39.3 | 38.8 |
| | SCOP-B (Ours) | 30.24 | 31.38 | 1.14 | 10.92 | 11.55 | 0.63 | 43.5 | **45.0** |
| Res34 | SFP(2018) [9] | 26.08 | 28.17 | 2.09 | 8.38 | 9.67 | 1.29 | 39.8 | 41.1 |
| | FPGM(2019) [10] | 26.08 | 27.46 | 1.38 | 8.38 | 8.87 | 0.49 | 39.8 | 41.1 |
| | Taylor (2019) [27] | 26.69 | 27.17 | 0.48 | N/A | N/A | N/A | 22.1 | 24.2 |
| | SCOP-A (Ours) | 26.69 | **27.07** | **0.38** | 8.58 | **8.80** | **0.22** | 39.7 | 39.1 |
| | SCOP-B (Ours) | 26.69 | 27.38 | 0.69 | 8.58 | 9.02 | 0.44 | **45.6** | **44.8** |
| Res50 | CP (2017) [11] | N/A | N/A | N/A | 7.80 | 9.20 | 1.40 | N/A | 50.0 |
| | ThiNet (2017) [26] | 27.12 | 27.96 | 0.84 | 8.86 | 9.33 | 0.47 | 33.72 | 36.8 |
| | SFP (2018) [9] | 23.85 | 25.39 | 1.54 | 7.13 | 7.94 | 0.81 | N/A | 41.8 |
| | Autopruner(2018) [25] | 23.85 | 25.24 | 1.39 | 7.13 | 7.85 | 0.72 | N/A | 48.7 |
| | FPGM (2019) [10] | 23.85 | 24.41 | 0.56 | 7.13 | 7.73 | 0.24 | 37.5 | 42.2 |
| | Taylor (2019) [27] | 23.82 | 25.50 | 1.68 | N/A | N/A | N/A | 44.5 | 44.9 |
| | C-SGD (2019) [4] | 24.67 | 25.07 | 0.40 | 7.44 | 7.73 | 0.29 | N/A | 46.2 |
| | GAL (2019) [23] | 23.85 | 28.05 | 4.20 | 7.13 | 9.06 | 1.93 | 16.9 | 43.0 |
| | RRBP (2019) [44] | 23.90 | 27.00 | 3.10 | 7.10 | 9.00 | 1.90 | N/A | 54.5 |
| | Hrank (2020) [22] | 23.85 | 25.02 | 1.17 | 7.13 | 7.67 | 0.51 | 36.7 | 43.7 |
| | PFP-A (2020) [21] | 23.87 | 24.09 | 0.22 | 7.13 | 7.19 | 0.06 | 18.1 | 10.8 |
| | PFP-B (2020) [21] | 23.87 | 24.79 | 0.92 | 7.13 | 7.57 | 0.45 | 30.1 | 44.0 |
| | SCOP-A (Ours) | 23.85 | **24.05** | **0.20** | 7.13 | **7.21** | **0.08** | 42.8 | 45.3 |
| | SCOP-B (Ours) | 23.85 | 24.74 | 0.89 | 7.13 | 7.47 | 0.34 | **51.8** | **54.6** |
| Res101 | SFP (2018) [9] | 22.63 | 22.49 | -0.14 | 6.44 | 6.29 | -0.20 | 38.8 | 42.2 |
| | FPGM (2019) [10] | 22.63 | 22.68 | 0.05 | 6.44 | 6.44 | 0.00 | 38.8 | 42.2 |
| | Taylor (2019) [27] | 22.63 | 22.65 | 0.02 | N/A | N/A | N/A | 30.2 | 39.7 |
| | PFP-A (2020) [21] | 22.63 | 23.22 | 0.59 | 6.45 | 6.74 | 0.29 | 33.0 | 29.4 |
| | PFP-B (2020) [21] | 22.63 | 23.57 | 0.94 | 6.45 | 6.89 | 0.44 | 50.4 | 45.1 |
| | SCOP-A (Ours) | 22.63 | **22.25** | **-0.32** | 6.44 | **6.16** | **-0.28** | 46.8 | 48.6 |
| | SCOP-B (Ours) | 22.63 | 22.64 | 0.01 | 6.44 | 6.43 | -0.01 | **57.8** | **60.2** |

error after reducing 50.0% FLOPs ). Even for the compact MobileNetV2, our method can still prune 40.3% FLOPs with only 0.24% accuracy drop.

## 4.2 Comparison on ImageNet

We further conduct extensive experiments on the large-scale ImageNet (ILSVRC-2012) dataset and compare the proposed method with SOTA filter pruning methods. The single view validation errors of the pruned networks are reported in Table 2. We prune the pre-trained networks with two different pruning rates, denoted as 'SCOP-A' and 'SCOP-B', respectively[5]. Compared with the existing criteria for filter importance (*e.g.*, SFP [9],FPGM [10],Taylor [27] and Hrank [22]), our method achieves the lower errors (*e.g.*, 24.74% top-1 error with 'SCOP-B' *v.s.* 25.50% with 'Taylor' on ResNet-50), while more FLOPs are pruned (*e.g.*, 54.6% *v.s.* 44.9%). It verifies that the proposed method with knockoff as the control group can excavate redundant filters more accurately from a pre-trained network. Compared with other SOTA filter pruning methods (*e.g.*, GAL[23], PFP [21]), our method also shows large superiority as shown in Table 2.

The realistic accelerations of the pruned networks are shown in Table 3, which are calculated by measuring the forward time on a NVIDIA-V100 GPU with a batch size of 128. Though the realistic

Table 3: Realistic acceleration ('Realistic Acl.') and theoretical acceleration ('Theoretical Acl.') on ImageNet.

| Model | Method | Realistic Acl. (%) | Theoretical Acl. (%) |
|---|---|---|---|
| Res18 | SCOP-A | 26.3 | 38.8 |
| | SCOP-B | 34.2 | 45.0 |
| Res34 | SCOP-A | 28.4 | 39.1 |
| | SCOP-B | 32.5 | 44.8 |
| Res50 | SCOP-A | 33.4 | 45.3 |
| | SCOP-B | 41.3 | 54.6 |
| Res101 | SCOP-A | 37.1 | 48.6 |
| | SCOP-B | 50.6 | 59.4 |

Table 4: Effectiveness of knockoff features as the control group. 'Error' denotes the error of pruned networks and 'Gap' is the increased error on CIFAR-10.

| Model | Method | Error (%) | Gap (%) | FLOPs ↓ (%) |
|---|---|---|---|---|
| Res32 | No control | 8.23 | 0.98 | 55.8 |
| | Noise | 8.15 | 0.90 | 55.8 |
| | Random sample | 8.22 | 0.97 | 55.8 |
| | Ours w/o bias | 7.86 | 0.56 | 55.8 |
| | Ours with bias | 7.78 | 0.53 | 55.8 |
| Res56 | No control | 6.83 | 0.53 | 56.0 |
| | Noise | 6.81 | 0.51 | 56.0 |
| | Random sample | 6.77 | 0.47 | 56.0 |
| | Ours w/o bias | 6.47 | 0.10 | 56.0 |
| | Ours with bias | 6.36 | 0.06 | 56.0 |

acceleration rates are lower than the theoretical values calculated by FLOPs due to the others factors such as I/O delays and BLAS libraries, the computation cost is still substantially reduced without any specific software or hardware.

### 4.3 Ablation studies

**Varying pruning rate.** The accuracies of the pruned networks *w.r.t.* the variety of pruning rates are shown in Figure 2. Increasing pruning rate means more filters are pruned and the parameters & FLOPs of the pruned network are reduced rapidly. Our method can recognize the redundancy of the pre-trained network, and the accuracy drop is negligible even pruning 50% of the parameters & FLOPs of ResNet-56.

**Effectiveness of knockoffs as the control group.** Knockoff features are used as the control group to minimize the influence of irrelevant factors, and their effectiveness is empirically investigated in Table 4. 'No control' denotes that no control group is used and filters are pruned only based on the scaling factors $\beta^l$ of real features. The result shows that test error is increased from 6.36% to 6.83%

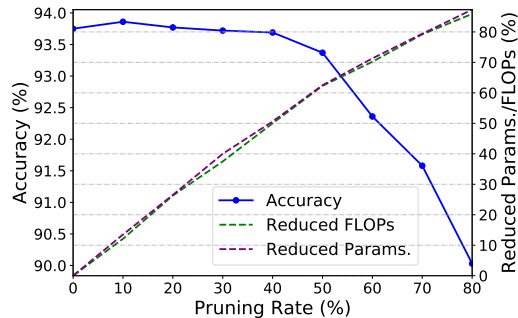

Figure 2: The accuracies and parameters/FLOPs of the pruned ResNet-56 vary *w.r.t.* pruning rate on CIFAR-10.

on ResNet-56. This verifies that the control group plays an important role to accurately excavate redundant filters.

Except for feeding knockoff data to the network to generate auxiliary features, other data may also be used as auxiliary data. 'Noise' denotes using random noise sampled from normal distribution as auxiliary data, and 'Random sample' produces auxiliary data by randomly sampling data from the original dataset. Compared with knockoff data, 'Noise' does not obey the exchangeability (Eq. (3)) while 'Random sample' contains information about targets. They both incur larger errors compared with our method, which shows the superior of knockoff data (features) acting as the control group. What's more, biases satisfying Eq. (5) are added to the output of convolutional layers to strictly satisfy the second-order knockoff condition from a theoretical perspective. As shown in Table 4, adding the biases or not both can achieve high accuracies.

**Visualization of knockoff data and features.** We intuitively show the real data/features with their knockoff counterparts in Figure 3. The real data contains information about targets (*e.g.*, fish) which is propagated to the intermediate feature. A well-behaved deep network should utilize these information adequately to recognize the targets. The generated knockoff data are similar to the real images, but there are no targets. Thus, they almost provide no information about labels. The frequency distribution histogram of real/knockoff features (Figure 3 (c)) shows that knockoffs ('orange') approximately

coincide with those of real features ('blue'). More visualization results are shown in the supplementary material.

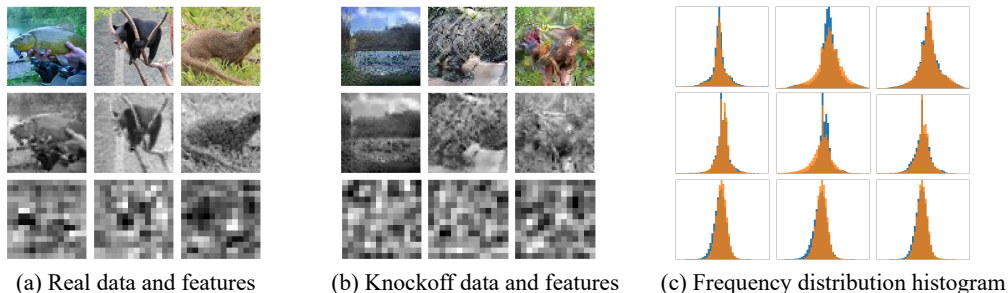

(a) Real data and features     (b) Knockoff data and features     (c) Frequency distribution histogram

Figure 3: Features of real data in ImageNet and their knockoffs. From top to bottom are input images, features in shallow layers and features in the deep layers. In (c), feature distributions of real data and knockoffs are marked in 'blue' and 'orange', respectively.

## 5   Conclusion

This paper proposes a novel network pruning method via scientific control (SCOP), which improves the reliability of neural network pruning by introducing knockoff features as the control group. Knockoff features are generated with the similar distribution to that of real features but contain no information about ground-truth labels, which reduces the disturbance of potential irrelevant factors. In the pruning phase, the importance of each filter is calculated according to the competition results of the two groups of features. In particular, filters that pay more attention to knockoff samples rather than real data will be removed for obtaining compact neural networks. Extensive experiments demonstrate that the proposed method can obtain better results over the state-of-the-art methods. For example, our method can reduce 57.8% parameters and 60.2% FLOPs of ResNet-101 with only 0.01% top-1 accuracy loss on ImageNet. In the future, we plan to research the design of scientific control in more deep learning problems, such as neural architecture search.

## Broader Impact

Network pruning is an effective model compression strategy to accelerate the inference of deep neural networks and reduce their memory requirement. It greatly promotes the deployment of deep neural networks on the massive edge devices such as mobile phones and wearable gadgets [24]. Even on a cheap device with limited computer capability, powerful models can still work well with the proposed pruning method. It lowers the barrier of the application of artificial intelligence and provides convenience to our works and lives.

## Funding Disclosure

This work is supported by National Natural Science Foundation of China under Grant No. 61876007 and Australian Research Council under Project DE-180101438.

## Footnotes

[1]The bias is omitted for symbol simplicity, which does not change the conclusion. Note that convolutional transformation is a special linear transform and can also be represented as this form.

[2] For architectures with BN layers, statistic $\boldsymbol{\mathcal{I}}^l$ is defined as $|\boldsymbol{\gamma}| \odot (\boldsymbol{\beta}^l - \tilde{\boldsymbol{\beta}}^l)$, where $\boldsymbol{\gamma}$ is the scales in BN layers and $|\cdot|$ denotes the absolute value operation.

[3] https://www.mindspore.cn/resources/hub

[4] https://www.mindspore.cn

[5]'SCOP-A' sets pruning rate as 30% and 'SCOP-B' as 35% on ResNet-18 and ResNet-34. On ResNet-50 and ResNet101, 'SCOP-A' sets pruning rate as 35% and 'SCOP-B' as 45%.

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
