[Supplementary Material]

# SCOP: Scientific Control for Reliable Neural Network Pruning (Supplementary Material)

**Yehui Tang**[1,2], **Yunhe Wang**[2], **Yixing Xu**[2], **Dacheng Tao**[3],
**Chunjing Xu**[2], **Chao Xu**[1], **Chang Xu**[3]
[1]Key Lab of Machine Perception (MOE), Dept. of Machine Intelligence, Peking University.
[2]Noah's Ark Lab, Huawei Technologies.
[3]School of Computer Science, Faculty of Engineering, University of Sydney.
yhtang@pku.edu.cn, {yunhe.wang, yixing.xu, xuchunjing}@huawei.com,
{dacheng.tao,c.xu}@sydney.edu.au, chaoxu@cis.pku.edu.cn.

## S1  Proof for Lemma 2

**Lemma 2** *Suppose $\tilde{v}^l$ is the second-order knockoff counterpart of input feature $v^l$, the corresponding output feature $\tilde{v}^{l+1} = t(\tilde{v}^l) + \tilde{b}^l$ is still the second-order knockoff of $v^{l+1} = t(v^l) + b^l$, where bias $b^l$ and $\tilde{b}^l$ are random variables with zero means, and the covariance matrix of the joint distribution $[b^l, \tilde{b}^l]$ satisfies:*

$$\text{cov}[b^l, \tilde{b}^l] = \begin{bmatrix} \Sigma_{b^l} & \Sigma_{b^l} + W^T \text{diag}\{s^l\}W - diag\{s^{l+1}\} \\ \Sigma_{b^l} + W^T diag\{s^l\}W - \text{diag}\{s^{l+1}\} & \Sigma_{b^l} \end{bmatrix}. \quad \text{(S.1)}$$

*where $b^l$, $\tilde{b}^l$ have the same covariance $\Sigma_{b^l}$ and $diag\{s^l\}$, $diag\{s^{l+1}\}$ are diagonal matrices.*

**Proof**  *Recall that $\tilde{v}^l$ is the second-order knockoff of $v^l$ meaning the expectation & covariance of $[v^l, \tilde{v}^l]$ and $[v^l, \tilde{v}^l]_{swap(\tilde{S})}$ are the same for arbitrary subset $\tilde{S}$, and $\tilde{v}^l \perp\!\!\!\perp Y | v^l$. The input $\tilde{v}^l$ contains no targets and thus the output $\tilde{v}^{l+1} = t(\tilde{v}^l) + \tilde{b}^l$ also contains no information about labels, which satisfies the independence condition (Eq. (4)). Their means are equal due to batch normalization [3], and hence we focus on checking whether the covariance matrices of $[v^l, \tilde{v}^l]$ and $[v^l, \tilde{v}^l]_{swap(\tilde{S})}$ match in the following. Actually, $[v^l, \tilde{v}^l]$ and $[v^l, \tilde{v}^l]_{swap(\tilde{S})}$ having the same covariance is equivalent to that the covariance of $[v^l, \tilde{v}^l]$ satisfies the following form, i.e.,*

$$\text{cov}[v^l, \tilde{v}^l] = \begin{bmatrix} \Sigma_{v^l} & \Sigma_{v^l} - \text{diag}\{s^l\} \\ \Sigma_{v^l} - \text{diag}\{s^l\} & \Sigma_{v^l} \end{bmatrix}, \quad \text{(S.2)}$$

*where $\Sigma_{v^l}$ is the covariance of $v^l$ and $diag\{s^l\}$ is any diagonal matrix making the covariance matrix positive semi-definite [1]. As $\tilde{v}^l$ is the second-order knockoff counterpart of $v^l$, Eq.(S.2) satisfies for $[v^l, \tilde{v}^l]$. Via simple calculation, the covariance matrix of output $[v^{l+1}, \tilde{v}^{l+1}]$ is:*

$$\text{cov}[v^{l+1}, \tilde{v}^{l+1}] = \begin{bmatrix} W^T \Sigma_{v^l} W + \Sigma_{b^l} & W^T \Sigma_{v^l} W + \Sigma_{b^l} - diag\{s^{l+1}\} \\ W^T \Sigma_{v^l} W + \Sigma_{b^l} - diag\{s^{l+1}\} & W^T \Sigma_{v^l} W + \Sigma_{b^l} \end{bmatrix}, \quad \text{(S.3)}$$

*Denoting $W^T \Sigma_{v^l} W + \Sigma_{b^l}$ as $\Sigma_{v^{l+1}}$, Eq. (S.3) is reformulated as:*

$$\text{cov}[v^{l+1}, \tilde{v}^{l+1}] = \begin{bmatrix} \Sigma_{v^{l+1}} & \Sigma_{v^{l+1}} - diag\{s^{l+1}\} \\ \Sigma_{v^{l+1}} - diag\{s^{l+1}\} & \Sigma_{v^{l+1}} \end{bmatrix}, \quad \text{(S.4)}$$

*which satisfies the form of Eq. (S.2), indicating that $[v^{l+1}, \tilde{v}^{l+1}]$ and $[v^{l+1}, \tilde{v}^{l+1}]_{swap(\tilde{S})}$ also have the same covariance. Thus, $\tilde{v}^{l+1}$ is still the second-order knockoff of $v^{l+1}$.*

|  (a) Input data. | (b) Features in shallow layers. | (c) Features in deep layers. |

Figure S1: Visualization of the distribution of features *w.r.t.* samples on ImageNet.

Figure S2: Knockoff data and features on ImageNet. From top to bottom are input images, features in shallow layers and features in the deep layers.

To ensure that biases $[\boldsymbol{b}^l, \tilde{\boldsymbol{b}}^l]$ exist, the covariance matrix in Eq. (S.1) needs to be positive semi-definite. Through standard Schur complement calculation, the semi-definite condition can be derived, *i.e.*, $\mathrm{cov}[\boldsymbol{b}^l, \tilde{\boldsymbol{b}}^l] \succeq 0$ if and only if Eq. (S.5) holds,

$$\mathrm{diag}\{s^{l+1}\} \succeq W^T \mathrm{diag}\{s^l\}W,$$
$$2\Sigma_{\boldsymbol{b}^l} \succeq \mathrm{diag}\{s^{l+1}\} - W^T \mathrm{diag}\{s^l\}W. \tag{S.5}$$

With $\boldsymbol{s}^l$ given by the previous layer, it is easy to satisfy Eq. (S.5) by appointing $\mathrm{diag}\{\boldsymbol{s}^{l+1}\}$ and $\Sigma_{\boldsymbol{b}^l}$. For example, the diagonal elements in $\mathrm{diag}\{\boldsymbol{s}^{l+1}\}$ can be set as the maximum eigenvalue of matrix $W^T \mathrm{diag}\{s^l\}W$ added by a small positive constant $\gamma$ and then $\Sigma_{\boldsymbol{b}^l}$ is set as $(\mathrm{diag}\{\boldsymbol{s}^{l+1}\} - W^T \mathrm{diag}\{s^l\}W)/2 + \gamma I$, where $I$ is the identify matrix. The diagonal matrix $\mathrm{diag}\{s^0\}$ in the input layer is calculated with the real and knockoff data. With the given mean and covariance, biases $[\boldsymbol{b}^l, \tilde{\boldsymbol{b}}^l]$ can be directly generate to modify the outputs of convolutional layers in the procedure of excavating redundant filters.

## S2    Details of generating knockoff data

We train a generative adversarial network [2] to construct knockoff data. The knockoff data are generated by the generator and then sent to the discriminator to verify whether the knockoff condition (Definition 1) holds. The generator and discriminator are optimized alternately and the loss function provided by [4] is adopted. Considering that images are high dimension data, we take multiple pixels as a whole when exchanging elements between real data and knockoff data. To reduce the training cost on ImageNet, knockoff data with size $64 \times 64$ are generated first and then upsampled to $224 \times 224$. For the network architectures, multiple convolutional (deconvolutional) layers are stacked to construct the generator and discriminator on CIFAR-10, and ReNet-like architectures are adopted on ImageNet. All the models are optimized with Adam [5] optimizer. On CIFAR-10, the learning rate, batchsize and number of iterations are set to 0.001,128 and 20K, which are set to 0.001, 512 and 30K on ImageNet.

## S3 More Visualization Results

The distribution of features *w.r.t.* samples are shown in Figure S1, and 10K samples are sampled from ImagNet dataset. The blue points denote the concatenation of real features and knockoff features, *i.e.*, $[\mathcal{A}^l, \tilde{\mathcal{A}}^l]$. The orange points are the features after swapping, *i.e.*, $[\mathcal{A}^l, \tilde{\mathcal{A}}^l]_{swap(\tilde{S})}$, which are obtained by swapping half of the elements in $\mathcal{A}^l$ and $\tilde{\mathcal{A}}^l$. Both the two features are mapped to the 2D space for intuitive visualization and each point denotes a sample. It shows that swapping elements between $\mathcal{A}^l$ and $\tilde{\mathcal{A}}^l$ does not have much affect of the distribution, which is accordant with the exchangeability property (Eq. (3)) of knockoffs. More knockoff data and features are shown in Figure S2, which intuitively show that no targets are included in the knockoff data and features.