[Reviews · NeurIPS 2020]

Review 1

Summary and Contributions: ====================Post Rebuttal========================= I have read the reviews of the fellow reviewers and the response of the authors. They have addressed the 2/3 of my points. Methodology and communication is somewhat lacking, and prevents a thorough understanding of the approach and implementation (acknowledged by R4 as well). The authors have agreed to work on this, although it is not sure how they plan to go about it. The results for the methods compared were changing between tables, and it was suggested to present mean and std. dev values, which the authors have now included. Finally, I believe that this new approach to compression, by the comparison of knock-off features to real features where the knock-off features are optimised to be conditionally independent of the label given real features, also opens up new ways of examining and analysing deep neural networks beyond compression. The visualisations of the knock-off features at the end of the paper are the first evidence of this. In the light of this, I would like to change my score to marginally above acceptance. However, the uptake of this work and its value to the community are really contingent upon properly blending of the causal literature with NN compression literature and needs to be presented accordingly. ======================================================= The paper presents an approach for filter pruning, whereby if the feature at the output of a particular filter-set is found to be redundant, then that filter-set can pruned away without losing accuracy, while making the network more compact. For identifying the redundant features, they compare the real features with the features obtained from knockoff data (called knockoff features). And, if the network relies more on a particular knockoff feature than on the corresponding real feature (reliance is measured in terms of the scaling factors \beta), then the real feature is picked for pruning.

Strengths: 1. The idea explored in the work is interesting, in that it tries to avoid the instantiation of arbitrary thresholds and hyperparameters for pruning. Instead, it compares each feature to its knockoff counterpart to pick the redundant ones. 2. The results demonstrate that the approach is sound, it is mostly at par with other approaches, and sometimes slightly ahead in terms of the compression achieved. 3. The research subject is relevant to NeurIPS in the capacity that more compact and high-performing networks have a huge real-world applicability for deep networks and machine learning at large.

Weaknesses: 1. The explanation could be made clearer. At many places in the methodology the authors lose the forest for the trees. Perhaps, it is because they skip explaining terms like 'knockoff' or control group and treatment group, which is not often found in pruning literature and needs to be motivated/explained in situ. 2. In the result tables, the accuracies of the 'original' architectures keep changing. They should perhaps be reported with mean and std. deviation so that the variance in the baseline doesn't obsure the performance comparison of different approaches. 3. The results with random samples and noise as knockoff data, as reported in Table 4 are particularly interesting. However, to me, they also seem to highlight that fact that similar compression rates can be obtained without much difference in accuracy (half a percent point on cifar10), by using simpler schemes to create 'knockoff feautures' rather than using a computational expensive generator to generate knockoff data. Maybe, more experiments could be done with larger architectures and datasets to explore this.

Correctness: Yes, they seem to be.

Clarity: Yes, in terms of language and comprehensibility.

Relation to Prior Work: Yes, but it can be improved. A dedicated related work section is missing. There are limited references and most are directly presented in the results table where one doesn't know how to interpret them without reading the references first.

Reproducibility: Yes

Additional Feedback: 1. The use of the term 'filter' is highly confusing. Particularly, in reference to Eq.1. When the authors say filters, do they mean a kxk kernel of which there are NxM different instances in a layer - N being # of input channels and M being # of output channels. Alternatively, is the filter word used for an Nxkxk tensor of which there are M different instances? [From the extension to Eq. 2, it seems to be the latter, since they associate each filter with a single output feature. Is this correct? Please clarify.] 2. Although the authors mention geometric median or norm-based pruning of filters (hand-crafted techniques requiring hyperparameter search), they haven't brought to light any learning based approach to pruning features (and thus, the associated filter sets). I would imagine that those may be computationally more efficient than the one currently proposed, and also make for a good baseline. Do the authors have any comments on this?


Review 2

Summary and Contributions: The paper proposes a method for (filter) pruning neural networks by introducing a 'scientific control' group of knockoff features. In contrast with previous works where knockoff features will have to be generated for each layer of the network, the paper proposes using knockoff input data and provides proof that the properties of these knockoff data/features are preserved across the layers of the deep neural network. To prune the networks, additional layers are added to pretrained networks (ResNet, etc.) to learn “control scales” for both true features and the knockoff features. Filters with large scales for knockoff features are subsequently pruned. Experiments were performed on CIFAR-10 and ImageNet with various network architecture and performance was compared to many other recent pruning techniques to demonstrate the superiority of the proposed scheme.

Strengths: Theoretical grounding/analysis - The paper provides sound theoretical claims for the approach with formal definition and requirements for the knockoff features. It also provides analysis of how such requirements remain preserved across the layers of a typical deep network (e.g., CNN) to justify the use of knockoff data as part of the input to the pruning process. Algorithmic novelty - While the ideas of knockoff features as well as filter pruning are not new, the proposed algorithm to use carefully crafted knockoff data for filter pruning seems novel to the best of my knowledge. Empirical evaluation - The experiments are pretty solid - with multiple data sets, multiple network architectures, comparison with competing methods - the results are quite convincing. The visualizations of knockoff data/features and study of varying pruning rates are also useful. Significance and relevance to the NeurIPS community - Network pruning has a lot of practical relevance for the NeurIPS community as many applications intend to deploy fast, agile deep nets on smaller hardware (low computation power) for real-time or near real-time operations. Therefore, effective methods to prune redundant parameters while maintaining performance will be very useful.

Weaknesses: 1. The paper has some clarity issues - the initial language of 'scales' (larger/smaller) for real/knockoff features being the criteria of pruning is quite confusing as the notion of scale is still not well established. While this becomes clear later on (mathematically), clearer contribution statements upfront will be useful. Similarly, there seems to be a suggestion that there is novelty in the generating process of the knockoff data, but the main paper does not discuss it (which it probably should, maybe a result table can go to Supp. to create space) and also the process of knockoff data generation lacks details even in the supplementary section. For example, in the SI, it is briefly explained that the discriminator verifies if the generated knockoff features satisfy Definition 1. How is this verified? My suggestion is that the authors should clearly articulate the novel contributions (and adoptions from previous works) in algorithm and analysis at the end of introduction. 2. The criteria for selecting which filters are important is explained clearly, but it is not very clear how it is done in practice. Since the knockoff data are passed in as input and propagated through every layer, are the filters pruned at every layer or only for a few layers? 3. The notion of "knockoff data contains no information about the target or the labels" is mathematically clear, but the concept should be better explained in words, perhaps with examples if possible.

Correctness: To my best understanding, the claims, methodology and empirical investigation approach are correct.

Clarity: The paper has clarity issues as laid out in the weakness comments.

Relation to Prior Work: The paper has multiple elements such as generating the knockoff data/features, the pruning algorithm. While it describes the relevant literature landscape well, more clarity on the exact differences and novel contributions would be helpful. See the weakness comments for further details.

Reproducibility: Yes

Additional Feedback: Is there any limitation or drawback of this algorithm? For example, is the control scale always dominating for either the real features or knockoff features in actual implementation? What happens if the layers produce control scales that are close to 0.5/0.5 for knockoff features and real features? Typos/grammar issues, e.g., Line 141: 'be approximated preserved across', Line 268: ground truth spelling error, Post-rebuttal: I have read the authors' response. The authors acknowledge the 'clarity' issues raised by myself and other reviewers. If accepted, the authors should address them in the final version (as they suggested in the rebuttal). I am not changing my scores.


Review 3

Summary and Contributions: This paper proposes a novel network pruning method by setting up a scientific control. Knockoff features are generated and then used as the control group, which reduces the disturbance of irrelevant factors. The authors also theoretically analyze the knockoff condition in the forward propagation of the network and derive knockoff features effectively given knockoff data. Redundant filters are discovered by comparing the real and knockoff features and then those redundant filters are pruned to get a compact network. Experiments results show that the proposed method suppresses the existing methods and ablation studies are also conducted to analyze the method.

Strengths: +This paper provides a new perspective to discover network redundancy and may inspire the community. Most of the existing network pruning methods only focus on the network itself and the original training data. While this paper tries to discover redundant filters by introducing extra information, knockoff features (data), to assist discover redundant filters. + The proposed scientific control is reasonable to improve the reliability of network pruning. Many irrelevant factors may disturb the judgement of filter importance. Some of them can be enumerated (e.g., fluctuation of input), while more factors are potential and cannot be named. With the scientific control, all those irrelevant factors are excluded together and thus redundant filters can be discovered reliably. + The theoretical analysis about knockoff data and features provides much convenience to obtain knockoff features in each network layer. With generated knockoff data, knockoff features in all layers can be derived effectively. + The experiment results are impressive. The proposed method is compared with many SOTA network pruning methods published recently, and shows significant superiority to them. Ablation studies are also conducted to verify the necessity of setting up the scientific control and the effectiveness of utilizing knockoffs as the control group. The visualization results intuitively reflect the property of knockoff data and features.

Weaknesses: Overall, the proposed method is novel and effective. The paper is also well written. However, I still have some minor concerns on this manuscript. - It will be clearer that replacing the art picture in Figure 1 with the actual generated knockoff images (Figure 3 (b)). The actually generated Knockoff data in Figure 3 (b) can intuitively reflect their property and placing them in the diagram (Figure 1) will help understanding. -Definition 1 describes the property of knockoff data, including exchangeability and independence. When describing the exchangeability property in Equation (3),‘swap()’denotes the element swapping operation between two features. The authors are suggested to provide a simple example illustrating how the swapping operation conducts. It helps readers to understand the definition of knockoff data.

Correctness: The claims, method and the empirical methodology are correct.

Clarity: This paper is well written. The motivation and details of the methods are clearly described.

Relation to Prior Work: The difference of this work between existing works are clearly discussed. The paper explores a new perspective to discover redundancy.

Reproducibility: Yes

Additional Feedback: The weaknesses mentioned above should be addressed.


Review 4

Summary and Contributions: #################Post-Rebuttal##################################### I have read the author's rebuttal, the authors have clarified a lot of missing details. Nonetheless, I still have some concerns about the theoretical details. First, the authors claim that Eq.(8) gives an approximation of Eq.(6). This is somewhat misleading. I think the goal here is not to let ||[A \trilde{A}]||_\le c. It is more like a elementwise selection between $A_l$ and $\tilde{A}_l$. So I think Eq.(6) and Eq.(8) do not agree with each other. Second, It is a bit disappointing to see that the authors only guarantee first and second-order moment matching for the knock-off filters instead of the entire distribution (Although the authors provide some practical evidence). It is then not for sure whether the 'control' group is 'clean' enough as it was claimed to be. Of course, I think this paper is interesting in some sense. But It seems that a significant re-organization (especially for the methodology) is still necessary before publication. I will keep my score as it is correspondingly. But I will not fight for a rejection. ################################################################# This paper proposes a filter pruning method by the design of knockoff data/features. The knockoff counterpart is random variables satisfying exchangeability and independence from ground truth labels. This paper incorporates knockoff counterpart as a scientific control group to help discover redundancy in real features. Then filters that produce more redundancy are pruned accordingly. Experimental results on real-world datasets demonstrate that the proposed method reduces model parameters and FLOPs significantly with little accuracy reduction.

Strengths: 1. Novelty. This paper proposes a novel method for determining the importance of filters to conduct filter pruning. The idea of incorporating knockoff features as a scientific control group to help discover redundant features is interesting. 2. The empirical experiments are well-designed and a substantial number of experiments are conducted. The results are promising and convincing to show the effectiveness of the proposed method.

Weaknesses: 1. The optimization algorithm is not clearly clarified. Since there is an L_0-norm constraint about the number of non-zero elements of features in each layer, the optimization is non-trivial and is expected to be clarified. 2. The specific design of feature selection layer is simple and this paper fails to dive deeper into the proposed control scales \beta^l and \tilde{\beta^l}. 3. Some notions are not clearly clarified or not eye-catching. For example, the selection layer (Eq.7) calculates A^{l+1} by mixing real features and knockoff features. Is there a need to calculate \tilde{A^{l+1}}, if so, how to calculate it? Also, I do not really catch the role of biases.

Correctness: The claims and method are reasonable and the authors give theoretical proof about the knockoff conditions. Empirical experiments are well-designed and substantial to convince the claims.

Clarity: It is roughly well-written. The overall logic and organization are good. However, some parts are kind of confusing, for example, for me, the role of bias b^l and \tilde{b^l} is not clear enough. Also, there are some grammar issues and typos. Part of them are listed as follows: 1. Line 56, ‘sate-of-the-art’ should be ‘state-of-the-art’. 2. Line 51, ‘two groups’ should be ‘two groups’. 3. Line 144, ‘be directly generate’ should be ‘be directly generated’. 4. Line 86, ‘two property’ should be ‘two properties’. 5. Line 91, ‘do not change’, it seems that ‘do’ does not match the subjective ‘swapping …’ 6. Line 170, ‘with constrain’ should be ‘with the constraint’

Relation to Prior Work: It seems that the contributions and distinctions from previous work are not clearly demonstrated.

Reproducibility: Yes

Additional Feedback: 1. The optimization algorithm or procedure is expected to be seen. As shown in Eq. 6, the constraint is in the form of L0-norm, thus is non-trivial to solve. 2. It is seen in Table 1 that the competing methods are different for ResNet20, ResNet32, ResNet-56 and MobileNetV2 on CIFAR-10. Similar problems happen in Table 2 on ImageNet. The explanation that why different methods are selected when the backbone model is different is expected to be seen. 3. The specific design of the feature selection layer is simple and this paper fails to dive deeper into the proposed control scales \beta^l and \tilde{\beta^l}. Will it happen that the model collapses into a trivial solution that \beta^l=1 and \tilde{\beta^l}=0?

[Author Response · NeurIPS 2020]

We sincerely thank the anonymous reviewers for their support and constructive comments.

**To Reviewer #1. Q1**: *Presentation.* **A**: Treatment group denotes the features to be researched while control group is the features for controlling irrelevant factors, which are instantiated as real and knockoff features respectively (Line 81-90 in the main paper). We will refine the presentation and add a dedicated section for revisiting related-works.

**Q2**: *Different results in the baseline.* **A**: We collect the baseline results from the published papers and their test errors of the 'original' networks are slightly different. We re-implement the competing methods under the same baseline, and the results of ResNet-56 on CIFAR-10 with mean/std are shown below. More results will be included in the final version.

| Method | Original Error (%) | Pruned Error (%) | Gap (%) | Params. ↓ (%) | FLOPs ↓ (%) |
|---|---|---|---|---|---|
| GAL (2019) [17] | $6.30 \pm 0.24$ | $6.91 \pm 0.14$ | 0.61 | 42.4 | 50.0 |
| SCP (Ours) | $6.30 \pm 0.24$ | $\mathbf{6.36 \pm 0.09}$ | **0.06** | **56.3** | **56.0** |

**Q3**: *Effectiveness of knockoffs.* **A**: Results of ResNet-50 on ImageNet using noise or knockoffs are shown below. Since ImageNet is more complex, *e.g.*, 1,000 categories and images of high resolution ($224 \times 224$), utilizing noise data is hard to obtain good results. More results will be included in the final version.

| Method | Error (%) | Gap (%) | FLOPs ↓ (%) |
|---|---|---|---|
| No control | 26.22 | 2.37 | 54.6 |
| Noise | 25.86 | 2.01 | 54.6 |
| Ours with bias | 24.74 | 0.89 | 54.6 |

**Q4**: *The term 'filter'.* **A**: There are $M$ filters in Eq. (1), and each of them is an $N \times k \times k$ tensor, where $k \times k$ is the kernel size (*e.g.*, $3 \times 3$) and $N$ is the number of input channels. We will refine the presentation around the definitions.

**Q5**: *Efficiency.* **A**: Learning based approaches (*e.g.*, CP [8], GAL [17]) are compared in Table 1 and 2 in the main paper. These methods re-train the original network to learn the importance of filters. Compared with them, our method fixes the network weights and only tunes the control scales when discovering redundant filters, which is more efficient. The practical consuming time of pruning ResNet-56 (no fine-tuning) on CIFAR-10 is shown below (A V100 GPU).

**To Reviewer #2. Q1**: *Clarity issues.* **A**: 1) The term 'scales' denotes the magnitudes of $\beta_l$ and $\tilde{\beta}_l$, which will be replaced with 'scaling

| Method | CP | GAP | Ours |
|---|---|---|---|
| Time (min) | 46 | 25 | 16 |

factors' and defined at the beginning of the paper. 2) We adopt similar loss function for the discriminator as Knockoff-gan [9] and propose to aggregate multiple elements for efficiently generating knockoff data, which will be fully described in the final version. The novel contributions will also be discussed at the end of introduction as your suggestion.

**Q2**: *Pruning procedure in practice.* **A**: Unimportant filters in each layer will be pruned. As discussed in the main body (Line 184-186), for an arbitrary convolutional layer, filters with small $(\beta_j^l - \tilde{\beta}_j^l)$ will be recognized as redundancy.

**Q3**: *Concept of knockoff data.* **A**: Specifically, knockoff data do not contain real objects of any category (*e.g.*, goldfish, snail) in the real dataset, as shown in Figure 3 of the main body. We will add more explanations in the final version.

**Q4**: *Limitation.* **A**: Thanks for this nice concern. 1) Control scales are distributed in range [0,1] as shown in Figure R1. 2) It does not matter when scales are close to 0.5/0.5, since we focus on sorting scales of different filters, rather than $\beta^l$ and $\tilde{\beta}^l$ of the same filter.

**To Reviewer #3. Q1**: *Figure 1.* **A**: We will replace the art picture in Fig. 1 with the actually generated knockoff data.

**Q2**: *Example for swapping operation.* **A**: Suppose that $\mathcal{A}$=[0.1,0.18,-0.1],$\tilde{\mathcal{A}}$=[0.13,0.16,-0.15], and then $[\mathcal{A}, \tilde{\mathcal{A}}]$=[0.1,0.18,-0.1,0.13,0.16,-0.15]. If $\tilde{S} = \{2\}$, the swapped feature $[\mathcal{A}, \tilde{\mathcal{A}}]_{swap(\tilde{S})}$=[0.1,0.16,-0.1,0.13,0.18,-0.15].

Figure R1: $\beta^l$ *w.r.t.* epoch.

**To Reviewer #4. Q1**: *Optimization procedure.* **A**: Eq. (6) is the general formulation for feature selection. In practice, we use Eq. (8) to avoid $\ell_0$-norm and then Adam optimizer can be applied.

**Q2**: $\beta^l, \tilde{\beta}^l$ *in the feature selection layer.* **A**: Thanks for this constructive comment. To have an explicit understanding, we illustrate the change of $\beta^l$ (*i.e.*, the first conv layer in last stage of ResNet-56 on CIFAR-10) during the optimization in Figure R1. Wherein, each curve denotes the control scale ($\beta_j^l \in \beta^l$) of a specific convolution filter. Similarly, $\tilde{\beta}^l = \mathbf{1} - \beta^l$ has the opposite phenomenon. Since most of existing deep neural networks are of heavy design for the accuracy reason, they will not collapse to $\beta^l = \mathbf{1}$. Visualizations on $\beta^l, \tilde{\beta}^l$ and corresponding discussions will be included in the final version.

**Q3**: *Notions.* **A**: 1) We need to calculate knockoff feature $\tilde{\mathcal{A}}^{l+1}$, which is the $l+1$-th layer's feature map of the network with knockoff data as input, *i.e.*, $\tilde{\mathcal{A}}^{l+1} = f^{l+1}(\tilde{X}, \mathcal{W}^{1:l+1})$. 2) Biases $b^l$ and $\tilde{b}^l$ are the modified terms from theoretical derivation, which ensure that the knockoff condition satisfies in the forward propagation of neural network from a theoretical perspective. These notions will be clarified more detailedly in the final version.

**Q4**: *Different methods and backbone models.* **A**: We apply the proposed method on different backbone models to verify its generalization ability. The results of competing methods on different backbones are collected from their original papers for fair comparison. These methods have their own experimental settings and lack results on some backbones. For example, Hrank [16] did not report results on ResNet20, ResNet32 and MobileNetV2. Thus we do not include it.

**Q5**: *Clarity.* **A**: Thanks, all of these typos and minor comments will be carefully fixed in the final version, and more discussions on the broader impact will be included.

[Meta-Review · NeurIPS 2020]

This paper presents a method to prune filters in convolutional neural networks by introducing a scientific control group of knockoff features to reduce the disturbance of irrelevant factors. The authors also analyze the knockoff condition theoretically and derive the knockoff features given the knockoff data. Experiments are performed on CIFAR-10 and ImageNet. The reviewers and AC have read the author feedback carefully in addition to all the reviews. It is generally agreed that the proposed method is novel and interesting in that there is no need to specify arbitrary thresholds and hyperparameters for pruning. The theoretical analysis is sound and provides the requirements for knockoff features. The experiment results show that the method is effective, either comparable to or better than the SOTA pruning methods. Nevertheless, the clarity of the paper has room for improvement to make it appeal better to the readers. One reviewer upgraded the overall score during the post-rebuttal discussion. At the end of the discussion period, three out of four reviewers are supportive of accepting this paper and the other reviewer does not object to its acceptance. Consequently, it is recommended by the AC to accept the paper for poster presentation. All reviewers have made specific suggestions that can help to improve the clarity and presentation of the paper. The authors are recommended to consider them seriously for the revision.